# OpenReview forum: "What Characterizes Effective Reasoning? Revisiting Length, Review, and Structure of CoT"
_ICLR.cc/2026/Conference — Submitted to ICLR 2026_

### Official Review · Reviewer_MNov · 2025-10-19

**Soundness:** 2
**Presentation:** 3
**Contribution:** 1
**Rating:** 2
**Confidence:** 4

**Summary:**

This paper aims to explore the factors that may affect the effective reasoning of LLMs. In specific, the authors mainly study on three factors" response length, review ratio, and failed-step fraction.  By conducting experiments on ten large reasoning models (LRMs) and two reasoning datasets, the authors highlight that: **given a same question, longer response length, higher review ratio, or higher failed-step fraction brings negative impact to the reasoning accuracy**. Then, the authors show that re-ranking and selecting responses based on the values of these three metrics can improve the reasoning accuracy, furthering validating the relationship between the metrics and reasoning performance.

**Strengths:**

1. The studied problem, effective reasoning, is important. The motivation to study the effects of response length, review ratio, and failed-step fraction is clear and straightforward.

2. The paper is well-written. The experimental analysis validates the claim of the paper.

**Weaknesses:**

1. **The contribution is very limited, and the findings do not provide new insights.** The potential negative effects of longer CoT length and more failed steps are already discovered in previous studies such as [1], the fact that selecting shorter responses under different samplings to a same problem leads to higher accuracy has also been revealed in [2]. The findings in this paper are not surprising to the community.

2. The proposed method that using failed-step fraction as an indicator to achieve more effective test-time scaling is equivalent as introducing an extra LLM critic. Introducing LLM critic or external reward models is a common practice on improve model's inference-time scaling performance [3,4].

3. The evaluated datasets are limited. The authors only conduct experiments on a subset of HARP problems and 198 GPQA-Diamond problems.


[1] Yang, Wenkai, et al. "Towards thinking-optimal scaling of test-time compute for llm reasoning." arxiv 2025

[2] Hassid, Michael, et al. "Don't Overthink it. Preferring Shorter Thinking Chains for Improved LLM Reasoning." arxiv 2025

[3] Madaan, Aman, et al. "Self-refine: Iterative refinement with self-feedback." NeurIPS 2023

[4] Gao, Bofei, et al. "Llm critics help catch bugs in mathematics: Towards a better mathematical verifier with natural language feedback." arxiv 2024

**Questions:**

Regarding the definition of review ratio in Line 202, why is the ratio calculated based on the proportion of all characters in the review chunk rather than the proportion of tokens?

---

> ### Author Response · Authors · 2025-11-22
> **Rebuttal 1/2**
>
> We thank the reviewer for the time and helpful comments.
>
> >The contribution is very limited, and the findings do not provide new insights. The potential negative effects of longer CoT length and more failed steps are already discovered in previous studies such as [1], the fact that selecting shorter responses under different samplings to a same problem leads to higher accuracy has also been revealed in [2]. The findings in this paper are not surprising to the community.
>
> We respectfully disagree that our contributions are limited or identical to [1] and [2]. While high-level intuitions about "shorter is better" exist in the community, scientific progress requires moving from anecdotal observation to rigorous, causal verification. Our work distinguishes itself through experimental setting (RL models at inference vs. SFT training), metric definition (exploration vs. error), and statistical rigor.
>
> **Distinction from [1] (RL models at inference vs. SFT training)**
>
> [1] approaches the "shorter is better" hypothesis by 1) comparing non-thinking vs. thinking models, and 2) training non-thinking models via SFT on imitation data of different lengths. In contrast, our work analyzes the inference-time dynamics of RL-trained thinking models. Specifically, we analyze intra-model dynamics: for a single model on a specific question, what distinguishes a correct reasoning path from an incorrect one? This paired approach rigorously controls for confounding factors arising from model training and intrinsic question difficulty—regimes that represent a fundamental departure from [1].
>
> [1] does look into erroneous steps. However, fail-step in our work is defined strictly differently:
> [1] measures correctness: They count reasoning rounds containing erroneous steps using an external judge (GPT 4o).
> We measure exploration: We extract the reasoning graph to identify "failed explorational steps" (steps that do not contribute to the final reasoning path), regardless of local correctness. By considering the fraction of failed explorational steps, we decouple the length effect from simple error rates, allowing for the rigorous causal testing that [1] does not perform.
>
> **Distinction from [2] (Statistical Power and Confounders)**
>
> While [2] is closer to our test-time setup, their study lacks statistical power and control.
>
> - Scale: [2] evaluates only 3 models on fewer than 100 data points. We evaluate 10 models (including frontier models) on a significantly larger dataset covering both math and science.
>
> - Rigor: [2] relies on grouped averages and simple shortest/longest comparisons, which are susceptible to noise and Simpson’s Paradox. They do not control for question difficulty.
>
> Our approach: We design a conditional correlation test to explicitly rule out confounding factors on the question side. We provide analysis across difficulty classes (based on human-expert annotation) and, crucially, establish causal interventions to explain why the negative correlation exists.
>
> In summary, we provide the **fine-grained, statistically controlled evaluation** that prior works lack. We validate the "shorter is better" intuition not merely as an observation, but through **difficulty-stratified analysis and causal mechanisms**. Furthermore, the reviewer overlooks our contributions beyond length: we explicitly examine the **review ratio** and the ((failed-explorational step fraction**. Neither of these factors is well addressed in prior work, and both offer significant new insights into thinking model behavior and its relationship with performance.
>
> >The proposed method that using failed-step fraction as an indicator to achieve more effective test-time scaling is equivalent as introducing an extra LLM critic. Introducing LLM critic or external reward models is a common practice on improve model's inference-time scaling performance [3,4].
>
> We clarify that this comparison stems from the misunderstanding addressed above: **our "failed-step" metric measures exploration structure, not correctness**.
>
> Standard LLM critics and reward models [3,4] function by judging semantic truth (correct vs. incorrect). In contrast, our metric identifies steps that are discarded during the reasoning graph construction (steps that do not contribute to the main reasoning path, such as explorational dead ends). This distinction is crucial because our metric remains well-defined even for incorrect CoT trajectories: it identifies "failed" explorational branches within a wrong CoT, where the main path is labeled "successful" despite being factually incorrect.  This distinction is already emphasized in Footnote 1 (Page 5) of the original submission. Therefore, our method is structurally and functionally distinct from correctness-based critic methods.

---

> > ### Author Response · Authors · 2025-11-22
> > **Rebuttal 2/2**
> >
> > >The evaluated datasets are limited. The authors only conduct experiments on a subset of HARP problems and 198 GPQA-Diamond problems.
> >
> >
> > We thank the reviewer for the comment. We believe the strength of our evaluation lies in the depth and density of the reasoning analysis rather than the raw count of problems. Unlike standard benchmarking, our study aims to characterize reasoning behavior while controlling confounding factors, which requires intensive sampling. In our evaluation, we examine 10 reasoning models. For each question, we generate 16 chains of thought (CoTs) per model for evaluation and 64 CoTs for test-time selection. Each reasoning trace contains over 5,000 tokens on average, amounting to more than 1.2 billion tokens in total. This extensive scale and diversity provide strong support for the universality and robustness of our claims.
> >
> > Furthermore, our datasets were specifically chosen to cover distinct reasoning modalities: mathematics (HARP) and science (GPQA). We observe consistent trends across both domains and all 10 model families. Given this high degree of stability across different tasks and models, we are confident that our conclusions are robust.
> >
> >
> > >Regarding the definition of review ratio in Line 202, why is the ratio calculated based on the proportion of all characters in the review chunk rather than the proportion of tokens?
> >
> > We selected character count to ensure strict comparability across the ten distinct models evaluated, since models employ different tokenizers with varying vocabularies and compression rates. While prior work like [1] attempts to standardize this by re-tokenizing all outputs with a single tokenizer, we hold that character count provides the most direct, model-agnostic measure of actual generation length.
> >
> > We thank the reviewer once again for the time and effort. Please let us know if we have addressed all the questions.

---

### Official Review · Reviewer_smJm · 2025-10-23

**Soundness:** 3
**Presentation:** 3
**Contribution:** 2
**Rating:** 4
**Confidence:** 4

**Summary:**

**Motivation**: Large reasoning models often produce very long chain-of-thought traces to gain accuracy, but what actually makes a good trace is unclear and long outputs raise cost and latency.  The paper revisits whether making thoughts longer or adding more review really helps, and asks if structural traits of the trace explain success across models and tasks.

**Approach**: The authors study ten models on math and science, measure two lexical metrics at the character level (Length and Review Ratio from LLM-labeled chunks), and extract a reasoning graph to compute the Failed-Step Fraction (FSF).

**Key Results**: Within the same question, shorter traces and lower Review Ratio are linked with higher accuracy, and FSF is the strongest and most consistent predictor across models and difficulty levels. For causal tests, picking answers by lowest FSF gives the largest pass@1 gains, up to about 10% on AIME, and removing failed branches raises continuation accuracy by roughly 8–14%.

**Strengths:**

**Motivation:** The paper tackles core questions about how LLMs reason. It states clear, testable questions and uses simple metrics at the character level and the graph level, which allows fair model comparisons while controlling for question difficulty.

**Evidence:** The evidence is strong and not just correlational. FSF predicts correctness across tasks, models, and difficulty bands; choosing traces with lower FSF boosts pass@1 more than length or review, and trimming failed branches improves continuation accuracy.

**Message:** The take-home message is practical. Do not default to very long chains of thought; use structure-aware selection and light pruning to get higher accuracy with lower cost and latency.

**Weaknesses:**

**Novelty of Finding:** The paper’s main claims feel close to earlier results about CoT length and review. Prior works (cited in the submitted paper) have already shown that adding “wait” can help at first but can also hurt, and that shorter traces can match or beat longer thinking, which raises the bar for what is new here. Also, regarding FSF and recovery, the paper itself notes that models often do not fully “unsee” early mistakes when they backtrack, and related work on self-verification probes hidden states without guaranteeing reliable recovery. This aligns with reports of over-verification effects and limited self-correction, which suggests the failure to recover can persist even with verification steps [1,2,3]. Clearer positioning against these threads would help.

[1] Training Language Models to Self-Correct via Reinforcement Learning

[2] Large Language Models Cannot Self-Correct Reasoning Yet

[3] When Can LLMs Actually Correct Their Own Mistakes? A Checklist

**Reliance on LLM-as-judge and extraction prompts:** Several steps depend on LLMs judging other LLMs. Review spans are labeled with the Llama 4 Maverick judge, and the paper shows a confusion matrix against human labels, but the approach could still inherit judge or prompt bias. Robustness checks like judge swaps, alternative prompts, and more human spot checks across tasks would strengthen the claims. Graph extraction also uses Claude 3.7, which introduces another source of dependency that would benefit from ablations or agreement analyses.

**CoT Intervention and missing baselines:** The strongest gains come from re-ranking many samples per question. The test-time selection setup generates 64 candidates per problem, then picks the top one, which is costly even if each trace is shorter. It would be useful to see guidance methods that push the model toward low-FSF traces during decoding, or lower-sample selectors with similar gains. Comparisons against structured search methods like Tree-of-Thoughts and common baselines such as self-consistency would also clarify cost-accuracy tradeoffs.

**Questions:**

1. How sensitive are FSF estimates to the specific graph extraction and failed-node labeling prompts? Would a different judge shift the results?
2. Can we reduce FSF during generation with lightweight controls, so we need fewer samples for selection?
3. Do these findings hold for tasks like coding, or tool use where long context and retries are common?

---

> ### Author Response · Authors · 2025-11-22
> **Rebuttal 1/3**
>
> We sincerely thank the reviewer for the detailed comments and for recognizing the strength of our evidence and our practical takeaways.
>
> > Novelty of Finding: The paper’s main claims feel close to earlier results about CoT length and review. Prior works (cited in the submitted paper) have already shown that adding “wait” can help at first but can also hurt, and that shorter traces can match or beat longer thinking, which raises the bar for what is new here.
>
> We thank the reviewer for this opportunity to better position our work. While we agree that prior observations regarding "wait" tokens and length exist, our work fills a critical gap in generalizability, statistical rigor, and metric depth that prior works (Wang et al. 2025, Ghosal et al. 2025, Wu et al. 2025 and more) do not address.
>
> 1. Scope: Specific Interventions vs. General Laws
> Prior works primarily analyze specific test-time scaling interventions (e.g., forcing length via "wait" tokens) on limited scopes—often restricting analysis to single model families (e.g., DeepSeek), smaller models (<8B parameters), or purely synthetic math tasks. In contrast, we perform a comprehensive evaluation across ten frontier models (including SOTA RL-trained reasoning models) on both math and science domains. We move beyond analyzing how a specific engineering trick (like "wait") affects performance, to uncovering the fundamental, intrinsic relationship between reasoning behaviors and correctness.
>
> 2. Rigor: Removing Confounder and Leveraging Causal Verification
> Previous findings often rely on aggregated trends where model capability and question difficulty are confounded. We employ a statistically controlled design: by conditioning on both the specific model and the specific question, we isolate the intra-model dynamics to determine what distinguishes a correct CoT from an incorrect one. We further provide causal tests for length and review behaviors, rather than just correlational observations.
>
> 3. Novel Metrics: Beyond Length
> Finally, our analysis extends well beyond length and review. We introduce and measure a suite of novel features, including reasoning graph metrics (failed-step fraction), entropy, progressiveness, and motivation analysis (detailed in Appendix C.4, D, and F.1). These metrics provide new insights into the "reasoning texture" of large models that prior length-focused studies have not touched.
>
> In short, while prior works identified phenomena related to length in restricted settings, we establish generalizable laws of reasoning dynamics. We achieve this through a broader scope (10 frontier models), rigorous causal testing, and the introduction of multi-dimensional metrics—graph structure, entropy, progressiveness, and motivation—that go significantly beyond simple length analysis.
>
> > Also, regarding FSF and recovery, the paper itself notes that models often do not fully “unsee” early mistakes when they backtrack, and related work on self-verification probes hidden states without guaranteeing reliable recovery. This aligns with reports of over-verification effects and limited self-correction, which suggests the failure to recover can persist even with verification steps [1,2,3]. Clearer positioning against these threads would help.
>
> We thank the reviewer for raising this comparison. However, we argue that the findings in [1,2,3] and papers on over-verification describe a fundamentally different mechanism than the one we study. We distinguish our work from [1,2,3] and over-verification based on the inference paradigm: Post-hoc Correction vs. Intrinsic Exploration.
>
> Works like [1,2,3] typically evaluate post-hoc self-correction, where a model (often a non-thinking model) generates a complete answer and is then prompted to verify or correct it in a separate inference pass (or API call). The consensus in these works is that models struggle to self-critique a "committed" answer. In contrast, our work analyzes intrinsic, online reasoning within a single reasoning path from RL-trained thinking models.
>
> We clarify that our Failed-Step Fraction (FSF) measures explorational dead ends. Here, the model is not criticizing a finished product; it is actively navigating a search space in a single chain. A "failed step" often represents a valid step that simply did not lead to the target (a strategic dead end), leading the model to backtrack and explore a new branch. The dynamics of generating a path are fundamentally distinct from reviewing a completed output. While "over-verification" describes redundant checks on an existing solution, FSF captures the necessary pruning of search branches before a solution is reached.
>
> In summary, our work shifts the focus to the reasoning process prior to solution convergence. This approach uncovers critical exploratory dynamics that are fundamentally distinct from the post-hoc self-correction or over-verification studied in prior work.

---

> ### Author Response · Authors · 2025-11-22
> **Rebuttal 2/3**
>
> >Reliance on LLM-as-judge and extraction prompts: Several steps depend on LLMs judging other LLMs. Review spans are labeled with the Llama 4 Maverick judge, and the paper shows a confusion matrix against human labels, but the approach could still inherit judge or prompt bias. Robustness checks like judge swaps, alternative prompts, and more human spot checks across tasks would strengthen the claims. Graph extraction also uses Claude 3.7, which introduces another source of dependency that would benefit from ablations or agreement analyses.
>
> >How sensitive are FSF estimates to the specific graph extraction and failed-node labeling prompts? Would a different judge shift the results?
>
> We thank the reviewer for this important question. While using LLMs as judges always carries some risk of bias, we believe our results are robust for the following reasons:
>
> 1. 90% Human Agreement on Review Spans: We explicitly validated our judge against human labels (see Appendix) and found a 90% agreement rate. This high level of concordance confirms that for review identification, the judge aligns very closely with human judgment.
>
> 2. Consistent Predictive Power: For the reasoning graph—which is notoriously difficult for humans to annotate—our strongest evidence is the consistency of the results. Across every model family we tested, the extracted features consistently predicted performance and showed causal validity. If our extraction was driven by prompt artifacts or judge bias rather than real reasoning signals, we would not see such a clear, stable trend across all models. The consistent predictive power supports that the method captures a fundamental reasoning signal rather than extraction artifacts.
>
> 3. Finally, To address concerns about graph extraction sensitivity, we performed the requested ablations by swapping the extractor/judge models. Since graph extraction requires specific capabilities (GraphViz generation, complex instruction following), we compared extraction results using GPT-4o against the original Claude 3.7 pipeline across CoTs from both DeepSeek-R1 and Claude 3.7. We have added the results to Appendix D.3 in blue. We observe a strong FSF correlation between the different judge setups (0.82 for DeepSeek-R1 CoTs and 0.66 for Claude 3.7 CoTs on the HARP set). This indicates that FSF estimates are robust to the specific choice of judge, provided the model has sufficient reasoning and coding capability.
>
> > CoT Intervention and missing baselines: The strongest gains come from re-ranking many samples per question. The test-time selection setup generates 64 candidates per problem, then picks the top one, which is costly even if each trace is shorter. It would be useful to see guidance methods that push the model toward low-FSF traces during decoding, or lower-sample selectors with similar gains. Comparisons against structured search methods like Tree-of-Thoughts and common baselines such as self-consistency would also clarify cost-accuracy tradeoffs.
>
> > Can we reduce FSF during generation with lightweight controls, so we need fewer samples for selection?
>
> We clarify that the test-time selection setup is **designed as a causality test, not a proposal for a cost-effective, production-ready inference method**. Our goal was to examine the causal impact of specific features (like FSF). To do this rigorously: We utilized 64 candidates specifically to stress test the selection and ensure statistical significance. This allows us to claim with high confidence that FSF is a causal feature, rather than a correlation artifact.
>
> Because this is a feature attribution analysis rather than a search algorithm proposal, comparing directly to structured search methods like Tree-of-Thoughts would introduce confounding variables (e.g., search dynamics) that obscure the design purposes. Our results show that under this rigorous control, FSF is a consistently stronger causal signal than trace length or review steps.

---

> > ### Author Response · Authors · 2025-11-22
> > **Rebuttal 3/3**
> >
> > >Do these findings hold for tasks like coding, or tool use where long context and retries are common?
> >
> > To evaluate robustness in domains requiring long-context and precise logic, we extended our analysis to the Computer Science and Engineering subsets of MMLU-Pro. While full coding or agentic evaluations are computationally prohibitive (requiring more than billions of tokens generated), these subsets serve as strong proxies for technical reasoning. The results are summarized in Appendix F.3 in blue.
> >
> > As detailed in Figure 13, we replicate the behaviors observed in our main experiments, though with interesting domain-specific nuances:
> >
> > - Computer Science: For the Qwen family, sequence Length remains negatively correlated with correctness. For Deepseek and Claude, FSF retains its strong negative correlation with correctness.
> >
> > - Engineering: In the Engineering subset, we observe a wider, more generalized trend where both Length and FSF consistently correlate negatively with performance.
> >
> > These results confirm that our findings also hold for general reasoning tasks.
> >
> > We thank the reviewer once again for the time and effort. Please let us know if we have addressed all the questions.

---

> > > ### Comment · Reviewer_smJm · 2025-11-25
> > > **Response to Authors**
> > >
> > > Thank you for the response. While I agree that it addresses most of the concerns raised in my reviews, I still feel that the novelty of the findings has not been fully resolved.
> > >
> > > The comparison between your work and the pitfalls of self-verification does alleviate part of my concern. However, I still find FSF itself not particularly surprising, especially given that Reviewer MNov pointed out that a similar idea has already appeared in the literature. I have also read your response to Reviewer MNov, and my impression is that the differences you highlight are not scientifically novel, but rather amount to a more technically rigorous study of CoT paths.
> > >
> > > For these reasons, I will keep my original scores.

---

> > > > ### Author Response · Authors · 2025-11-27
> > > >
> > > > We thank the reviewer for their engagement.
> > > >
> > > > We respectfully seek further clarification on why FSF is viewed as analogous to an LLM critic or external reward feedback.
> > > >
> > > > As established in the literature (including the references provided by Reviewer MNov), the presence of a critic or external reward feedback improves model performance (**a positive correlation**). If FSF were acting as a critic, we would expect higher FSF to correlate with better outcomes. However, our results show that FSF **negatively** correlates with performance. This inverse relationship strongly suggests that FSF captures a phenomenon inherently different from—and perhaps opposite to—the beneficial guidance of a standard LLM critic.

---

### Official Review · Reviewer_7m35 · 2025-10-26

**Soundness:** 3
**Presentation:** 3
**Contribution:** 3
**Rating:** 6
**Confidence:** 4

**Summary:**

The authors introduce a graph view of Chain-of-Thought (CoT) to extract its structure and identify a single statistic—the Failed-Step Fraction (FSF), defined as the fraction of steps in abandoned branches—that consistently outperforms length and review ratio in predicting correctness across various models. To probe causality, they design two interventions. First, they rank candidate CoTs by each metric at test time, finding that FSF yields the largest pass@1 gains. Second, they edit CoTs to remove failed branches, which significantly improves accuracy, indicating that these failed branches bias subsequent reasoning. Taken together, these results characterize effective CoTs as those that fail less and support structure-aware test-time scaling over indiscriminately generating long CoTs.

**Strengths:**

The authors' methodology is logically sound, supported by ample experimentation, and the conclusions drawn contribute to enhancing model performance.

**Weaknesses:**

I acknowledge that the conclusions drawn by the authors are valuable, hence the positive score. However, approaches based on Prompting and Chain-of-Thought (CoT) do not fundamentally enhance the capabilities of LLMs. The authors' approach is akin to reviewing model outputs and If this idea could be integrated into the model's reinforcement learning training or data construction process, it would further enhance the value of this work.

**Questions:**

Same as above

---

> ### Author Response · Authors · 2025-11-22
>
> We thank the reviewer for their time and for finding that our paper offers valuable conclusions supported by ample experiments.
>
> > However, approaches based on Prompting and Chain-of-Thought (CoT) do not fundamentally enhance the capabilities of LLMs. The authors' approach is akin to reviewing model outputs and If this idea could be integrated into the model's reinforcement learning training or data construction process, it would further enhance the value of this work.
>
> We thank the reviewer for their valuable feedback.
>
> We respectfully wish to clarify the core contribution. Our work moves beyond simple prompting to understand what characteristics define effective reasoning at test-time. We provide strong, empirical insights into this question:
> - We find that effective reasoning on hard problems is characterized by more concise paths with less redundant 'review'.
> - Crucially, we show that failed exploration paths bias and degrade further reasoning.
>
> These findings suggest that effective test-time inference is not about maximizing token count, but about structured test-time search and active context management. We believe this contribution—understanding these features of effective reasoning—is significant in its own right. These insights could help the community design the next generation of more principled and efficient test-time schemes.
>
> We strongly agree with the reviewer that integrating these insights into the training process is a critical and exciting future direction. As the reviewer notes, this is a substantial endeavor. It would require a large-scale experimental setup to systematically design new reward-shaping mechanisms or analyze the complex effects of data mixtures on behavior for different base models. This is a non-trivial undertaking that we believe constitutes a full, separate investigation, which we are excited to pursue in follow-up work.
>
> We thank the reviewer once again for the time and effort.

---

### Meta-Review · Area_Chair_THT7 · 2026-01-04

**Summary:**

This paper aims to seek the factors that affect the effective reasoning of LLMs. Besides token length and review steps, this paper proposes a new metric, Failed-Step Fraction (FSF), to evaluate the efficiency of LLM reasoning. By analyzing reasoning samples and conducting statistical evaluations, the authors argue that token length and review steps are insufficient for reasoning quality examination, where FSF is more reliable. Moreover, the work also demonstrates the application superiority of FSF specifically in test-time inference selection.

While the reviewers acknowledge the authors' efforts in statistical analysis and experimental discussion, the consensus is that the paper does not meet the acceptance criteria due to the following key concerns:
1. **Lack of Novelty**: All reviewers noted that the paper fails to demonstrate a qualitative leap from existing work. The findings regarding reasoning length and review processes overlap significantly with prior literature. Furthermore, the utility of FSF is somewhat expected, and it fails to demonstrate a fundamental distinction from the LLM-as-a-Judge paradigm.
2. **Limited Contribution**: The paper's contribution is largely restricted to test-time inference selection. It lacks an exploration of FSF's potential impact during the training (SFT/RL) phase. It remains an unresolved challenge to devise training strategies that intrinsically incentivize the generation of low-FSF reasoning.

**Reviewer Concerns:**

- **Addressed Concerns**: The authors have successfully addressed specific technical inquiries in the rebuttal. This includes Reviewer msJm's concerns regarding the LLM-as-a-Judge reliance and experimental settings (W2, W3, Q1, Q3), as well as Reviewer MNov's questions concerning the test datasets and external LLM critic (W3, W2).
- **Outstanding Concerns**: However, the authors have not sufficiently articulated distinct, novel insights regarding data attribution that differentiate this work from prior studies. Furthermore, the practical application of FSF to model training or efficient inference sampling lacks clear elaboration. As a result, the primary concerns shared by the reviewers (7m35-W, smJm-W1, smJm-Q2, MNov-W1) have not been adequately resolved.

**Reviewer Scores:**

- For reviewer `7m35`: the score will be unchanged
- For reviewer `smJm`: the score will be unchanged
- For reviewer `MNov`: the score will be unchanged

---

### Decision · Program_Chairs · 2026-01-26

Reject